# Stakeholders' views and perspectives on treatments of visceral leishmaniasis and their outcomes in HIV-coinfected patients in East Africa and South-East Asia: A mixed methods study

**Joanne Khabsa**[1], **Saurabh Jain**[2], **Amena El-Harakeh**[1], **Cynthia Rizkallah**[3], **Dhruv K. Pandey**[4], **Nigus Manaye**[5], **Gladys Honein-AbouHaidar**[3], **Christine Halleux**[6], **Daniel Argaw Dagne**[2], **Elie A. Akl**[7,8]*

**1** Clinical Research Institute, American University of Beirut Medical Center, Beirut, Lebanon, **2** Department of Control of Neglected Tropical Diseases, World Health Organization, Geneva, Switzerland, **3** Hariri School of Nursing, American University of Beirut, Beirut, Lebanon, **4** Kala-azar Elimination Programme, World Health Organization Country Office, New Delhi, India, **5** Neglected Tropical Diseases, World Health Organization Country Office, Addis Ababa, Ethiopia, **6** Special Programme for Research and Training in Tropical Diseases (TDR), World Health Organization, Geneva, Switzerland, **7** Department of Internal Medicine, American University of Beirut, Beirut, Lebanon, **8** Department of Health Research Methods, Evidence, and Impact, McMaster University, Hamilton, Ontario, Canada

* ea32@aub.edu.lb

## Abstract

### Background

In visceral leishmaniasis (VL) patients coinfected with human immunodeficiency virus (HIV), combination therapy (liposomal amphotericin B infusion and oral miltefosine) is being considered as an alternative to liposomal amphotericin B monotherapy. We aimed to assess the views of stakeholders in relation to these treatment options.

### Methodology

In a mixed methods study, we surveyed and interviewed patients, government functionaries, programme managers, health service providers, nongovernmental organizations, researchers, and World Health Organization (WHO) personnel. We used the Evidence to Decision (EtD) framework for data collection planning and analysis. Constructs of interest included valuation of outcomes, impact on equity, feasibility and acceptability of the treatment options, implementation considerations, monitoring and evaluation, and research priorities.

### Principal findings/Conclusion

Mortality and non-serious adverse events were rated as "critical" by respectively the highest (61%) and lowest percentages (47%) of survey participants. Participants viewed clinical cure as essential for patients to regain productivity. Non-patient stakeholders emphasized the importance of "sustained" clinical cure. For most survey participants, combination

**Data Availability Statement:** All relevant data are within the manuscript and its Supporting Information files.

**Funding:** The authors received no specific funding for this work.

**Competing interests:** The authors have declared that no competing interests exist.

therapy, compared with monotherapy, would increase health equity (40%), and be more acceptable (79%) and feasible (57%). Interviews revealed that combination therapy was more feasible and acceptable than monotherapy when associated with a shorter duration of hospitalization. The findings of the interviews provided insight into those of the survey. When choosing between alternative options, providers should consider the outcomes that matter to patients as well as the impact on equity, feasibility, and acceptability of the options.

## Author summary

In East Africa and South Asia, the number of patients with visceral leishmaniasis (VL) co-infected with human immunodeficiency virus (HIV) has been increasing over the years. In addition to independently posing major health challenges, the two conditions have detrimental effects on each other. In light of new evidence on treatment regimens for this patient population, the World Health Organization (WHO) recently updated its evidence-based region-specific treatment recommendations. To inform this process, we assessed the views of stakeholders on the outcomes of interest and on medication treatment options. The mixed methods study consisted of an online survey and semi-structured interviews. Outcomes such as mortality, complications, clinical cure, relapse and serious adverse events were viewed as important because of the burden associated with their experience, their consequences, and the co-infection status of the patients. Outcomes such as non-serious adverse events were viewed as less important for reasons relating to patient tolerability and ease of treatment. As compared to monotherapy, combination therapy was found to be more equitable, acceptable, and feasible. While our findings are important to consider by clinicians treating patients with VL patients coinfected with HIV, they also inform decisions made by other stakeholders such as guideline developers and program managers.

## Introduction

Visceral leishmaniasis (VL) is an opportunistic infection in individuals infected with human immunodeficiency virus (HIV) [1]. Initially reported from southern Europe in the mid-1980s, *Leishmania* and HIV coinfections have progressively been reported from 45 countries worldwide [2]. In countries reporting fully on HIV status in VL patients (positive, negative or status unknown), the proportion of coinfections has increased (1.6%, 2.6% and 3.8% in 2016, 2017 and 2018, respectively) [3]. In 2020, <1% of VL cases were reported to be co-infected in the African Region (AFR), 12.3% in the Region of the Americas (AMR) and 2.8% in the South-East Asia Region (SEAR) [4]. These proportions vary from country to country depending upon the extent of HIV screening in VL patients. For example, in India, the percentage of Kala-azar (KA) cases tested for HIV increased from 47% in 2015 to 98% in 2020, representing an increase in the proportion of KA–HIV co-infected cases from 0.5% in 2015 to 4.1% in 2020 [5].

Both HIV/AIDS and VL pose clinical, diagnostic, therapeutic, and public health challenges as they are mutually reinforcing conditions with synergistic detrimental effects on each other. Immunosuppression caused by AIDS reduces the response to VL treatment, with lower cure rates, higher drug toxicities, higher relapses, and higher mortality rates than in immunocompetent patients. With each episode of relapse, therapeutic response becomes challenging with decreased relapse-free interval. While data on the impact of HIV–VL coinfection on patients'

quality of life are scarce, the perception is that the impact is devastating [6]. While timely VL treatment is essential for reducing morbidity and mortality of these patients, adequate management of HIV is also important (e.g., HIV testing, selection of antiretroviral therapy (ART), timing of ART and careful consideration of drug-drug interactions while selecting appropriate regimen), contributing most importantly to reducing risk of VL relapse.

Treatment of VL in HIV–coinfected patients is constrained by reduced numbers of therapeutic options due to few clinical trials, ineffectiveness, or drug toxicity. Until recently, clinical studies were available only on *L. infantum* from Mediterranean countries, with no data on *L. donovani* from Africa or Asia. This is a major challenge considering that *L. infantum* has different drug susceptibilities and less virulence than *L. donovani* and that relapses due to L. donovani occur at a lower CD4+ count. While currently the World Health Organization (WHO) recommends monotherapy of liposomal amphotericin B infusion at a dose of 3–5 mg/kg daily or intermittently for 10 doses (days 1–5, 10, 17, 24, 31 and 38) up to a total dose of 40 mg/kg as the first-line treatment [1], new evidence on combination regimens has become available from clinical trials recruiting HIV–VL coinfected patients in East Africa [7] and South-East Asia [8]. The regimens consist of combination of liposomal amphotericin B infusion (at a dose of 5 mg/kg on days 1, 3, 5, 7, 9 and 11, up to a total dose of 30 mg/kg body weight) and oral miltefosine (100 mg in divided doses for 14 days in South-East Asia and 28 days in East Africa respectively). Generally, in this patient population, the duration of admission or further treatment depends upon a case-by-case basis and upon a variety of factors such as delay in knowing HIV status, delay in seeking VL treatment, other comorbidities and opportunistic infections.

The emerging evidence on combination regimens has created the need for WHO to propose evidence-based region-specific treatment recommendations. The WHO guideline process follows the Grading of Recommendations Assessment, Development and Evaluation (GRADE) methodology that uses the Evidence to Decision (EtD) framework to structure the development of recommendations [9]. The EtD framework addresses both the evidence for the comparative effects of treatment options under consideration, and evidence about contextual factors. These factors include the valuation of the outcomes of interest, as well as the impact on equity, feasibility and acceptability of the treatment options [10, 11].

The overall objective of this study was to assess the views of stakeholders in relation to the medication treatment options of VL in HIV-coinfected patients (combination therapy of liposomal amphotericin B infusion and oral miltefosine, and monotherapy of infusion of liposomal amphotericin B). To meet that objective, we followed a mixed methods study design. Our findings were intended to help inform the panel judgements about the factors depicted in the EtD framework for the above-mentioned WHO recommendations [10, 11].

The specific objectives by study design were:

1. Quantitative design: to assess stakeholders' rating of the importance of outcomes, and their views on equity, feasibility, and acceptability;

2. Qualitative design: to explore the reasons underlying stakeholders' rating and views;

3. Mixed methods design: to integrate the findings from the quantitative and qualitative components of the study.

## Methods

### Ethics statement

We obtained ethical approval from the Research Ethics Review Committee (ERC) at WHO, the local Institutional Review Board at the University of Gondar in Ethiopia and the Indian

Council of Medical Research (ICMR)-Rajendra Memorial Research Institute of Medical Sciences (RMRI) in India. We obtained written informed consent from survey participants and oral consent from interview participants.

## Overall design

Following an a priori protocol that was peer-reviewed independently by two subject experts, we used a mixed methods approach that included a cross-sectional survey and semi-structured interviews. We collected the data between February and July 2020. We followed the Good Reporting of A Mixed Methods Study (GRAMMS) guidelines [12].

## Survey study

**Study population and sampling.** Participants were from VL-endemic countries in East Africa and South-East Asia, and included national neglected tropical disease officers, national ministry of health functionaries, subnational programme managers, district and referral hospital physicians and nurses, nongovernmental organizations, academic researchers, respective regional and country office WHO personnel, and members of WHO's South-East Asia Regional Technical Advisory Group for VL. We employed a convenience sampling methodology, by sending invitation emails to relevant listservs (computerized list of names and e-mail addresses). We obtained email addresses of stakeholders from databases available within WHO headquarter office, and contacted WHO regional offices for the African Region, Eastern Mediterranean Region and the Region for South-East Asia. The Ministry of Health of India, and WHO partners like Médecins Sans Frontières (MSF), Drugs for Neglected Diseases Initiative (DNDi) and Leishmaniasis East Africa Platform (LEAP) were also contacted. Based on obtained email addresses (n = 260), the majority of invited participants were from South-East Asia (n = 197, 76%). Those who accepted the invitation received the consent form and the online survey.

**Data collection.** We developed the survey tool based on questionnaires previously developed for other WHO guidelines [13–16], and pre-tested it before sending to potential participants. The questionnaire covered participant demographics, valuation of the outcomes of interest, and judgements related to impact on equity, acceptability, and feasibility of the two interventions being compared (S1 Appendix). It also included open-ended questions on implementation considerations, monitoring and evaluation, and research priorities. The latter were similar to those asked during the semi-structured interviews (see below). We asked participants to rate the importance of the outcomes of interest on a 9-point Likert scale. For the remaining measures, we used the response options provided in the EtD. We sent two reminder emails after 7 and 13 days respectively. We used DataForm online tool from LimeSurvey. The landing page started with a consent form; participation was voluntary and participants could decline or exit the survey at any stage. Data were collected anonymously and independently by a data manager.

**Data analysis.** We categorized the responses to the valuation of outcomes into critical (values of 7, 8 and 9), important (values of 4, 5 and 6), and of limited importance (1, 2 and 3). We used Stata (version 13) to conduct descriptive analyses. We analysed responses to open-ended questions along with the data from the semi-structured interviews.

## Semi-structured interviews

**Study population and sampling.** We included the same stakeholder groups as in the survey study (referred to as "non-patient stakeholders"), as well as VL patients with HIV-coinfection. For the non-patient stakeholders, we invited those who had responded positively to a

question about their willingness to participate in a semi-structured interview. The lead investigator (JK) conducted the interviews with non-patient stakeholders after obtaining informed oral consent. At the treatment centre in Ethiopia where the clinical trial was conducted [7], physicians attempted to contact all the patients who had participated in the trial, while ensuring voluntary participation, confidentiality, and anonymity. A similar process was followed in India [8], except that a random sample of patients was contacted. Wherever possible, attempts were made to contact all VL patients under trial by the treating physicians. Patients were given information about this study and were invited for the interview. Local principal investigators (DP and NM) then contacted patients who volunteered and expressed their willingness, obtained their informed oral consent and conducted the interviews.

**Data collection.** We developed an interview guide and discussed it with interviewers to maximize consistency. It covered the same constructs as the questionnaire but sought deeper views. Interviews with non-patient stakeholders were conducted in English, while interviews with patients were conducted in local language, using simplified language. We interviewed patients over the telephone, and non-patient stakeholders using an online meeting platform, and audio recorded all interviews. We stopped recruiting patients when we reached data saturation. We anonymized all transcripts using a unique identifier per interviewee.

**Data analysis.** We used the framework thematic analytical approach guided by the EtD framework, which consisted of the following seven stages [17]: (1) transcription of the recordings verbatim (AEH and JK). Since interviews with patients were conducted in local language, the local investigators (DP and NM) translated interviews to English and transcribed them at the same time; (2) familiarization with the transcripts (AEH, CR and JK); (3) data indexing, which consisted of labeling each meaningful datum, initially applied to a few transcripts using the predetermined constructs of EtD framework (AEH, CR and JK). Texts that did not fit the predetermined constructs were also labeled; (4) discussion of labels leading to the creation of categories, forming the basis for the analytical framework (AEH, CR, JK and GHA); (5) application of the analytical framework to all transcripts, adding new labels that did not fit into the existing analytical framework when encountered (AEH, CR and JK); (6) charting and examining allocation of subcategories for each category, and reexamination of the labels that did not fit the existing analytical framework (AEH, CR and JK); (7) comparison and contrasting of categories and subcategories to finalize the results (AEH, CR, JK and GHA). All coders (AEH, CR and JK) first coded the same interview as part of a calibration exercise. The rest of the interviews were coded by one person. Coders met to determine the analytical framework, and compare and contrast findings. At the end of the process, we generated a complete narrative of the findings, supplemented with quotes from individual interviews.

## Mixed methods integration

We used a convergent mixed methods research design (Fig 1) [18]. At the methods level, we used the survey as a gateway to sampling non-patient stakeholders. In addition, when collecting data, we used the same constructs for the survey and interviews. We compared and merged findings from the quantitative and qualitative analysis, and further analysed the findings for concordance, discordance or expansion. At the interpretation and reporting level, we followed the contiguous approach to integration [18].

## Results

### Online questionnaire survey

Out of 260 invitations, we received 177 complete responses (overall response rate = 68%, response rate in South-East Asia = 72%, response rate in East Africa = 51%). The collected

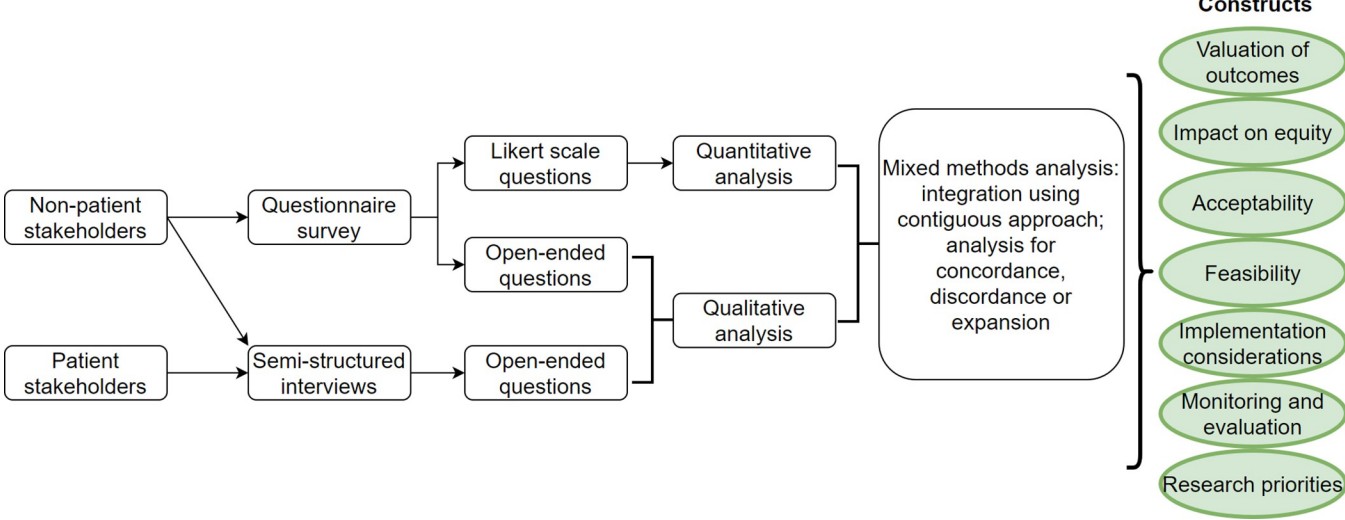

**Fig 1. Mixed methods design.**

data is presented in S2 Appendix. Most participants were males (n = 148, 85%), aged 31–50 years (n = 115, 65%), from South-East Asia (n = 142, 81%), and with a masters, doctoral or medical degree (n = 130, 74%). They belonged most commonly to two stakeholder groups: health providers (n = 62, 35%) and national neglected tropical diseases officers or subnational programme managers (n = 52, 30%). Half of the participants belonged to nongovernmental organizations (n = 88, 50%) (Table 1). S3 Appendix presents survey participants' characteristics per region.

## Valuation of the outcomes of interest

The outcome rated by the highest percentage of participants as "critical" was mortality (n = 105, 61%). This was followed by clinical cure at 6 months and clinical cure at the time of treatment completion (n = 99, 58%), relapse (n = 97, 57%), serious adverse events (n = 97, 57%), patient satisfaction (n = 97, 57%), and disease complications (n = 92, 54%). The outcome rated as "critical" by the lowest percentage of participants was non-serious adverse events (n = 81, 47%) (Fig 2).

**Impact on equity, acceptability, and feasibility of treatment alternatives.** When comparing combination therapy with monotherapy, the majority of participants (n = 105, 60%) responded that providing combination therapy would result in increased health equity (n = 70, 40% "increased", n = 35, 20% "probably increased"). Most participants (n = 139, 79%) responded that combination therapy is more acceptable (n = 97, 55% "more acceptable", n = 42, 24% "probably more acceptable"), and that combination therapy is more feasible than monotherapy (n = 100, 57%; n = 58, 33% "more feasible", n = 42, 24% "probably more feasible") (Fig 3).

## Semi-structured interviews

We interviewed 19 patients (nine from Ethiopia and 10 from India), and 10 non-patient stakeholders (two from Ethiopia and eight from India). Eight patients were from the combination therapy arm, and 11 were from the monotherapy arm. Non-patient stakeholders included four members of nongovernmental organizations, three physicians, two programme managers, and

**Table 1. Survey participants' characteristics (N = 177).**

|  | n (%)* |
|---|---|
| **Gender** |  |
| Female | 26 (15) |
| Male | 148 (85) |
| **Age (years)** |  |
| 18−30 | 23 (13) |
| 31−50 | 115 (65) |
| 51−64 | 31 (18) |
| > 64 | 8 (5) |
| **Region**** |  |
| South-East Asia | 142 (81) |
| East Africa | 32 (18) |
| Other | 1 (1) |
| **Highest attained educational degree** |  |
| Doctoral degree | 65 (37) |
| Master's degree | 65 (37) |
| Bachelor's degree | 35 (20) |
| Certificate or diploma | 4 (2) |
| None of the above | 7 (4) |
| **Stakeholder group** |  |
| Health providers/clinical officers | 62 (35) |
| National NTD officers/subnational programme managers | 52 (29) |
| Researchers | 25 (14) |
| Nongovernmental organizations | 14 (8) |
| Regional and country office WHO staff | 5 (3) |
| Policy-makers | 3 (2) |
| Members of WHO's Regional Technical Advisory Group on VL | 2 (1) |
| National ministry of health staff | 1 (1) |
| Not specified | 13 (7) |
| **Affiliation** |  |
| Nongovernmental organization | 88 (50) |
| Governmental organization | 48 (27) |
| International intergovernmental organization | 21 (12) |
| Academic institution | 16 (9) |
| Private for-profit organization | 2 (1) |
| Not specified/other | 2 (1) |

*Numbers do not always add to 177 due to missing data. Percentages represent valid percentages.

**Total of 14 countries

Abbreviations: NTD: neglected tropical diseases; WHO: World Health Organization; VL: visceral leishmaniasis

one representative of a national programme. Patients were mostly males (n = 17), and belonged to the following age groups: 18−30 years (n = 4), 31−50 years (n = 10), 51−64 years (n = 5). S2 Appendix provides the collected data, and S4 Appendix provides a detailed tabular presentation of the qualitative findings organized by construct with supporting quotes. The findings are summarized below.

**Valuation of the outcomes of interest.** Table A in S4 Appendix presents findings about the valuation for the following outcomes: mortality, disease-related complications, clinical

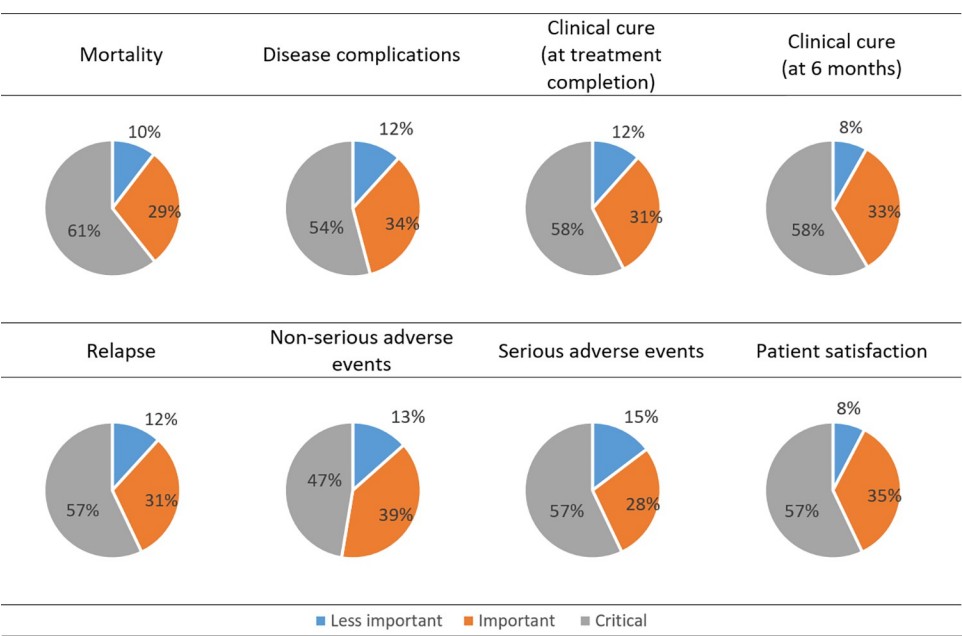

**Fig 2. Valuation of each of the outcomes of interest.**

cure, relapse, non-serious side effects, serious side effects, and patient satisfaction. The valuation of an outcome was driven by both the experience of the outcome itself (e.g., burden of relapse on the patient) and its consequences (e.g., relapse leading to mortality). For certain outcomes, it was also driven by the coinfection status of the patients (i.e., outcomes being more important in this specific population than in mono-infected or any patients).

Both patient and non-patient stakeholders considered reducing mortality as an important outcome. While non-patient stakeholders highlighted death as a critical outcome for any disease, they also noted the importance of mortality as an outcome in this specific population, since coinfection and immunosuppression place them at increased vulnerability. Patients considered survival as a *"miracle"* (India, P7), and recounted experiences of death among close relationships. Non-patient stakeholders were highly concerned about disease-related complications, for the possibility of being irreversible, being life-threatening, and having financial impacts. All participants perceived clinical cure as an opportunity for patients to resume their daily activities and be socially and economically productive. Non-patient stakeholders additionally placed a higher value on clinical cure at 6 months compared with clinical cure at the time of treatment completion, due to the increased risk of relapse among these patients.

Non-patient stakeholders described relapse as *"the main feature of HIV-VL"* (India, NP8) and *"the biggest challenge"* (India, NP5). This was due to how commonly relapse occurs, the risk of recurrent relapses, the resultant patient suffering and patients' doubts about treatment effectiveness, the impact on patients' mental health, and the ensuing risk of mortality. Relapse was also perceived to have negative effects from the health provider and community perspectives. From the community perspective, relapse could contribute to spreading the infection to other individuals, as patients are an *"important reservoir for the infection"* (India, NP2).

Non-patient stakeholders valued non-serious side effects less than the other outcomes, especially as compared with serious side effects. They considered non-serious side effects to be expected, well tolerated and easily treated, whereas serious side effects were viewed to result in additional burden on patients, and increased risk for permanent damage and mortality. Also,

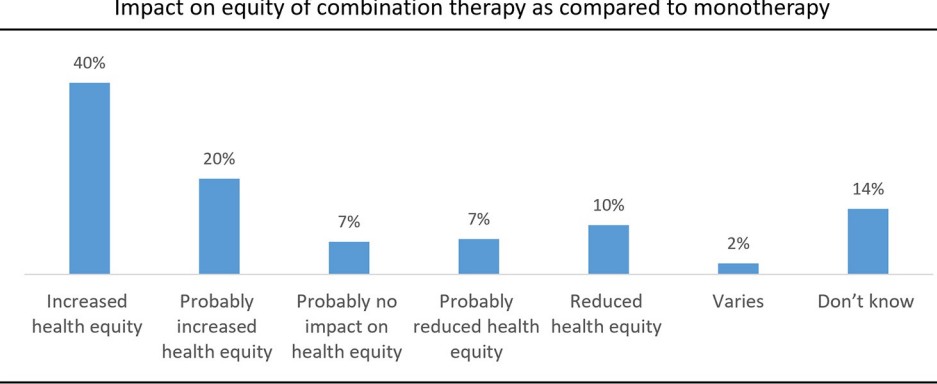

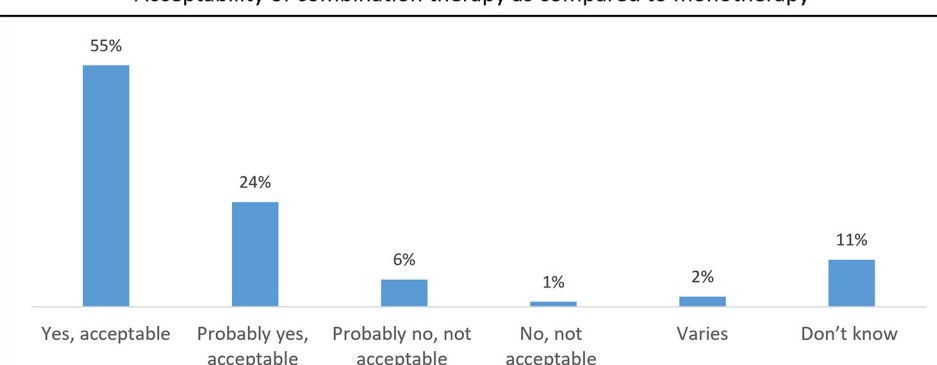

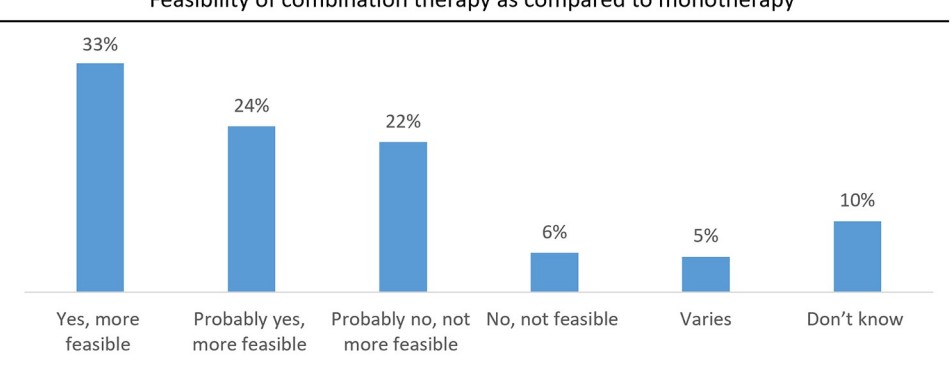

**Fig 3. Impact on equity, acceptability, and feasibility of combination therapy as compared with monotherapy.**

non-patient stakeholders were concerned about the risk of additive, more serious side effects, since this patient population is often receiving treatment for other diseases.

Participants rated patient satisfaction as an important outcome, for any patient and any disease. However, one non-patient stakeholder noted that satisfaction is usually overlooked: *"we do not look at it normally"* (India, NP7).

**Impact on equity.**   Most participants perceived no differential effect on equity between treatment alternatives as both are currently available free of charge and require hospitalization. They also noted that combination therapy is equally accessible for different geographical, gender and age groups. Nevertheless, few participants were concerned that pregnant women and women of childbearing age who may not prevent pregnancy were ineligible to receive

combination therapy (particularly administration of miltefosine in this category of patients), and that it may negatively impact equity.

In addition, a number of views on general equity issues relevant to patients with HIV–VL coinfection emerged (see Table B in S4 Appendix for additional details). To note that half of the patient participants from India reported initially receiving care at a private health facility for substantial amounts of money (range: US$ 133–2600) before receiving care at a government health facility.

**Feasibility.**   Longer duration of hospitalization was generally perceived to be less feasible due to the economic burden on the health system, the patients and their caregivers. Overall, there were no concerns for adherence issues with combination therapy, due to the treatment being completed at the hospital. One feasibility issue with combination therapy was the added difficulty of counselling-related contraindication in pregnant women (Table C in S4 Appendix,).

**Acceptability.**   Participants considered combination therapy to be more acceptable than monotherapy, when associated with a shorter duration of hospitalization. Both patient and non-patient stakeholders highlighted the burden of a longer duration of hospitalization on patients and their families, in terms of lost wages and need for accompaniment. In addition, some patients mentioned that they were less comfortable at the hospital than at home, that longer duration of hospitalization resulted in separation from their families and in inability to undertake daily tasks. Nonetheless, few participants preferred longer duration of hospitalization due to medical supervision available at the hospital and poor living conditions at home. The only two female participants noted that use of contraception was not an issue (see Table D in S4 Appendix for additional details).

**Implementation considerations, monitoring and evaluation, and research priorities.**
Participants reported on a number of implementation considerations, including but not limited to access to care, discrimination and stigma, patient involvement, compassion by health providers, counselling, and holistic care. Monitoring and evaluation suggestions included the parameters (e.g., adverse events, drug interactions) and logistics (e.g., tools, checklists) of monitoring. Research priorities included disease epidemiology, diagnostic aspects, prevention, disease progression and prognosis, need for additional data, treatment-related issues, immunology, relationship with HIV, and health systems issues. More detailed findings are provided in S5 Appendix for the survey study and Table E in S4 Appendix for the semi-structured interviews.

**Integrated findings.**   For the valuation of outcomes of interest, the qualitative and quantitative findings were generally concordant, in terms of all treatment outcomes being important to a certain extent. However, the qualitative findings had higher discriminatory ability than the quantitative findings. Indeed, while there were very small differences in the survey ratings of the different outcomes, the qualitative findings showed that some outcomes (such as mortality and relapse) were more valued that the rest of the outcomes (such as non-serious adverse events). One area of expansion was through qualitative findings providing justifications for valuation of outcomes.

For the contextual factors, quantitative and qualitative findings were also concordant, with combination therapy being more acceptable and feasible. Qualitative findings expanded on the reasons behind the ratings resulting from the quantitative study. Of note, 28% of participants indicated that combination therapy was "probably less feasible" or "less feasible". The qualitative study did not provide further insight into these findings.

## Discussion

### Summary of findings

We conducted a mixed methods study consisting of a cross-sectional survey and semi-structured interviews to assess the views of stakeholders as they relate to the treatment of VL in

HIV- coinfected patients. Mortality and non-serious adverse events were rated as "critical" by respectively the highest and lowest percentages of participants. Most survey participants perceived that combination therapy, compared with monotherapy, would increase health equity, is more acceptable, and is more feasible. The findings of the interviews were concordant with those of the survey, and provided important insight into the survey results. The one equity issue raised is related to the contraindication of combination therapy in pregnancy. Combination therapy associated with a shorter duration of hospitalization had higher feasibility and acceptability.

## Strengths and limitations

To our knowledge, this is the first study assessing the valuation of treatment outcomes and contextual factors in relation to treatment alternatives for VL patients with HIV. The mixed methods approach allowed us to maximize the respective benefits of quantitative and qualitative studies. While the quantitative survey study allowed us to reach a diverse sample of stakeholders with considerable sample size, the qualitative study provided further insight into their views. The qualitative study also allowed us to include patients with HIV–VL coinfection who would have had difficulties responding to the online survey.

We employed different strategies to enhance the rigour of our qualitative approach at the levels of sampling, data collection, analysis and reporting. These included recording of interviews, using the survey as a gateway to non-patient stakeholder participants, having the same person conducting the interviews with non-patient stakeholders being involved in the analysis (JK), discussions among the team, triangulation of data from patient and non-patient stakeholders, and objectivity in reporting (i.e. reporting of details which were not very commonly addressed). The validity of our mixed methods approach was enhanced by multiple-level integration (as mentioned above).

We included multiple types of stakeholders to reflect alternative perspectives. In fact, research on valuation of the outcomes by different stakeholders' groups often shows some differences in views among these groups [19, 20].

Using the EtD as the basis for the development of the survey and interview guide and as our analytical framework allowed us to tailor the findings to the specific recommendations under consideration by WHO. While collecting information for EtD purposes, we used a comparative approach (e.g., comparing the effects of combination therapy and monotherapy on equity). In addition, we addressed aspects related to the condition and its management in general (e.g., equity in relation to access to treatment).

One limitation of the study is that some patients were not very open about their views, even after probing. This could be due to patients being uncomfortable with discussion due to stigma and discrimination associated with their condition [6]. We are uncertain whether the use of face-to-face interviews would have yielded more in-depth information. However, they were not feasible given that this study was conducted during the COVID-19 pandemic. Another limitation is that this study had a smaller number of participants from East Africa as compared to South-East Asia. This was mainly due to unavailability of email addresses of health personnel in East Africa, and a lower response rate from East Africa as compared to South-East Asia. However, no obvious differences in perceptions between the two settings were noted. It is also interesting to note that in Ethiopia, some patients provided false phone numbers, while in India some of the phone numbers obtained from the hospital records were inaccurate. Finally, if the duration of the combination in practice will be longer compared to the trial protocols (e.g., due to repeated regimens), our findings need to be interpreted accordingly.

## Comparison with similar research

Nair et al. conducted a qualitative study to explore factors influencing the quality of life of patients coinfected with HIV and VL in India. They found that the most important indicators of "good quality of life" were income and livelihood [6]. This is consistent with our finding that the opportunity for the patient's earlier return to productivity was one of the drivers behind increased acceptability and feasibility of combination therapy. Another highly valued outcome was permanent cure [6], consistent with the views of non-patient stakeholders in this study. In addition, patients included in this study seemed to value resolution of symptoms higher than side effects, which is consistent with the literature on other conditions [21, 22].

Also consistent with our findings, Nair et al. found that patients in India reported high out-of-pocket expenditures when accessing treatment in the private sector [6]. Barriers to accessing medical care in the government system reported by participants in that study included long queues, poor quality of medicines, poor attendance by health professionals, and poor overall care. However, patients believed that they would achieve early cure through private care and preferred to seek private care for illnesses that they believed to be minor in nature [6].

## Implications for practice

When counselling patients on treatment alternatives, explanation of benefits and harms associated with the respective treatments should address the outcomes that matter to patients. Furthermore, when choosing one alternative over the other, health providers should address potential acceptability and feasibility concerns of patients.

Some themes such as general equity issues and implementation considerations are also important when implementing a therapeutic regimen. One has to ensure a holistic approach to patient care, addressing psychosocial, economic, and treatment needs. This is further confirmed by evidence showing that income, social support, and nutritional status are predictors for improvement in most quality of life domains after receiving treatment for patients with HIV, with and without VL [23].

Access to care remains paramount, given the geographical obstacles and importance of specialized care in treatment centres and provision of treatment free of charge. This could be facilitated by increasing awareness about free treatment among these patients, and addressing barriers to appropriate care in the governmental health sector. Given the concerns for adverse events and additive toxicity in combination therapy, pharmacovigilance systems also need to be strengthened.

## Implications for future research

The survey part of this study did not allow adequate discrimination between the different ratings, which has also been observed in previous similar surveys [13, 15, 16]. More research is needed to enhance the discriminatory ability of the survey instruments being used, and ensure its optimal use as part of mixed methods studies.

While the qualitative findings allowed us to capture the views of stakeholders on important contextual factors, future research should collect factual data on some of these factors. One example is field data on feasibility of administering one regimen versus others. Collecting this type of data would help further understand the influence of contextual factors on healthcare quality and decision-making. Also, funding agencies as well as research groups should consider the research topics suggested by participants as priorities. More generally, there is a need for studies assessing the impact of new approaches and interventions on quality of life in the population of interest.

## Supporting information

**S1 Appendix. Survey questionnaire.**
(DOCX)

**S2 Appendix. Survey and interview data.**
(XLS)

**S3 Appendix. Survey participants' characteristics by region.**
(DOCX)

**S4 Appendix. Detailed presentation of the findings from the semi-structured interviews organized by construct.**
(DOCX)

**S5 Appendix. Results of survey questions about implementation considerations, monitoring and evaluation, and research priorities.**
(DOCX)

## Acknowledgments

The authors thank Dr. Piero Olliaro and Dr. Axel Kroeger for their valuable inputs on the study protocol, and the Institutional Ethical Committees at the University of Gondar, Ethiopia, and the Internal Ethics Committee at ICMR–Rajendra Memorial Research Institute of Medical Sciences (RMRI), India, for providing ethical approval in spite of COVID-19 challenges. The authors also thank Dr. Rezika Mohammed Yesuf and Dr. Turid Piening for their contribution during the conduct of patient interviews in Ethiopia, and Dr. Pradeep Das and Dr. Krishna Pandey for their support in the data collection process and interviews of patients in India.

## Consent for publication Obtained from WHO

The authors alone are responsible for the views expressed in this article and they do not necessarily represent the views, decisions or policies of the institutions with which they are affiliated.

## Author Contributions

**Conceptualization:** Joanne Khabsa, Saurabh Jain, Christine Halleux, Daniel Argaw Dagne, Elie A. Akl.

**Data curation:** Joanne Khabsa, Amena El-Harakeh, Dhruv K. Pandey, Nigus Manaye.

**Formal analysis:** Joanne Khabsa, Saurabh Jain, Amena El-Harakeh, Cynthia Rizkallah, Gladys Honein-AbouHaidar, Elie A. Akl.

**Methodology:** Joanne Khabsa, Saurabh Jain, Christine Halleux, Daniel Argaw Dagne, Elie A. Akl.

**Project administration:** Saurabh Jain.

**Supervision:** Elie A. Akl.

**Writing – original draft:** Joanne Khabsa, Saurabh Jain, Elie A. Akl.

**Writing – review & editing:** Joanne Khabsa, Saurabh Jain, Amena El-Harakeh, Cynthia Rizkallah, Dhruv K. Pandey, Nigus Manaye, Gladys Honein-AbouHaidar, Christine Halleux, Daniel Argaw Dagne, Elie A. Akl.

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
