## [Decision Letter · Decision Letter 0]

12 Nov 2021

Dear Dr. Akl,

Thank you very much for submitting your manuscript "Stakeholders’ views and perspectives on treatments of visceral leishmaniasis and their outcomes in HIV-coinfected patients in East Africa and South-East Asia: a mixed methods study" for consideration at PLOS Neglected Tropical Diseases. As with all papers reviewed by the journal, your manuscript was reviewed by members of the editorial board and by several independent reviewers. In light of the reviews (below this email), we would like to invite the resubmission of a significantly-revised version that takes into account the reviewers' comments. 

We cannot make any decision about publication until we have seen the revised manuscript and your response to the reviewers' comments. Your revised manuscript is also likely to be sent to reviewers for further evaluation.

Sincerely,

Mitali Chatterjee

Associate Editor

Helen Price

Deputy Editor

Reviewer's Responses to Questions

**Key Review Criteria Required for Acceptance?**

**Methods**

-Are the objectives of the study clearly articulated with a clear testable hypothesis stated?

-Is the study design appropriate to address the stated objectives?

-Is the population clearly described and appropriate for the hypothesis being tested?

-Is the sample size sufficient to ensure adequate power to address the hypothesis being tested?

-Were correct statistical analysis used to support conclusions?

-Are there concerns about ethical or regulatory requirements being met?

Reviewer #1: The aim of the study is unclear. The design looks ok. A mixed study design and a variety of study participants were included. The conclusions could be expected to vary depending on the category of study participant. The sample size for qualitative study is ok. There are no ethical concerns.

Reviewer #2: Are the objectives of the study clearly articulated with a clear testable hypothesis stated?

Yes

Is the study design appropriate to address the stated objectives?

Yes

-Is the population clearly described and appropriate for the hypothesis being tested?

Yes

-Is the sample size sufficient to ensure adequate power to address the hypothesis being tested?

Yes

-Were correct statistical analysis used to support conclusions?

Yes

-Are there concerns about ethical or regulatory requirements being met.

Yes

Reviewer #3: Although the approach is interesting, I have a number of thoughts on the study. My main concerns are on the lack of participation from the East African setting, where the case numbers, challenges and level of expertise in VL-HIV is far higher than in the Indian setting where it is a relatively new phenomenon that is being treated/managed by a very limited group of stakeholders. Secondly, one of the most important parts of this study is the patient involvement. The numbers of patients interviewed (n=19) are really impressive and should have been enough to generate a wealth of evidence in what is one of the most under-described perspectives (I am assuming here that these were VL-HIV patients). 

However, the description of the methodology and subsequently the content and richness of the feedback presented in the manuscript does not seem to be adequately covered – recognising that the interviews were done over the phone – but this needs better description and explanation of what transpired and why this was not enough to get some clear thematics.

**Results**

-Does the analysis presented match the analysis plan?

-Are the results clearly and completely presented?

-Are the figures (Tables, Images) of sufficient quality for clarity?

Reviewer #1: The figures particularly figure 3 should be revised.

Reviewer #2: Does the analysis presented match the analysis plan?

Yes

-Are the results clearly and completely presented?

Yes

-Are the figures (Tables, Images) of sufficient quality for clarity?

Yes

Reviewer #3: Introduction:

• Note that atypical disseminated leish is defined as a stage 4 defining illness rather than VL.

• For example, in India, the percentage of Kala-azar (KA) cases tested for HIV increased from 47% in 2015 to 98% in 2020, representing an increase in the proportion of KA–HIV co-infected cases from 0.5% in 2015 to 4.1% in 2020 (4). 

- I’m unsure about this reference, but this number on proportions tested in India is very wrong. In 2015, barely any VL patients were being tested for HIV (certainly <10%). And 98% testing rates in 2020…its highly unlikely. The limitations of KAMIS need to be at least alluded to (eg there is no zero-reporting for HIV testing).

• This is a major challenge considering that L. infantum has different drug susceptibilities and less virulence than L. donovani and occurs at a lower CD4+ count.

- Unclear what is meant by ‘occurs at a lower CD4 count’? Both infections occur at low CD4 counts.

• The regimens consist of combination of liposomal amphotericin B infusion (at a dose of 5 mg/kg on days 1,3,5,7,9 and 11, up to a total dose of 30 mg/kg body weight) and oral miltefosine (100 mg in divided doses for 14 days in South-East Asia and 28 days in East Africa respectively)

- Note that the East African regimen is required to be repeated in many cases, so treatment duration is doubled. This should be checked and worded accordingly.

Methods:

• It is worth explaining what a listserv is; most people will not know this

• Were all interviews conducted in English/in person? Who coded the transcripts and were these double coded/QA’d by another researcher? There needs to be more robust description of the qualitative methodology in general. I see this is in appendix 2 = the authors should move this into the main manuscript or dedicate more space to describe this in the text.

• More details on the interview method/framework with the patients is needed

Results:

• Did the response split between Asia/Africa reflect the numbers of invitations sent? I.e what proportion were sent to participants from both regions? It’s a very Asia heavy participation, which is odd since the bulk of VL-HIV patients come from the African setting, as does the majority of evidence. Would be important to explain this.

• I think it could be important to split the table so that we can see the characteristics of the participants from Asia and Africa separately. Particularly for Africa, this will allow the reader to contextualise the feedback.

• Table 1: Region: 175/177 are reported – where are the other 2 from? Some of the strata do not add to 177, please check and annotate for missing data (either in the table or in the legend)

• Semi structured interviews with patients – specify if these were VL-HIV patients, and which treatments they had been treated with previously

**Conclusions**

-Are the conclusions supported by the data presented?

-Are the limitations of analysis clearly described?

-Do the authors discuss how these data can be helpful to advance our understanding of the topic under study?

-Is public health relevance addressed?

Reviewer #1: Limitations of the study are mentioned. The authors have not officially made specific conclusions and recommendations. However there is a hidden conclusion that for me appears to be unsubstantiated.

Reviewer #2: Are the conclusions supported by the data presented?

Yes

-Are the limitations of analysis clearly described?

Yes

-Do the authors discuss how these data can be helpful to advance our understanding of the topic under study?

Yes

-Is public health relevance addressed?

Yes partly.

Reviewer #3: • The one equity issue raised is related to the contraindication of combination therapy in pregnancy.

Worth re-checking this phraseology – combo treatment is contraindicated in pregnancy, lactation and in women of childbearing age unwilling or unable to take/use contraception

• The patient interviews over the phone are mentioned only at this point, this should be in the methodology section. How were patients selected and invited to participate (also worth mentioning how they were consented). Were their interviews recorded? More details are needed for this component as this is one of the most interesting parts of the study – beneficiary impact perception is critical, but we need to better understand how this was done.

• Where we had insufficient information from the patients’ perspective, we indicated this as part of the above findings..

Is unclear what the authors refer to here, there is no mention of this in the previous text.

• Limitations – it seems that the biggest limitation here is that the African region was massively under-represented; this needs much more discussion on the causes and reasons, as otherwise the study is really only valid for the Indian/South Asian setting

• Again the listserv – it is worth describing which listservs the authors refer to and from where they were acquired. I am surprised that snowball sampling was not conducted – is there a reason why? The VL-HIV world is quite small, and this could have been a very good way of identifying stakeholders...

• India phone numbers not being accurate = this is not surprising as people change sim cards very frequently in India.

**Editorial and Data Presentation Modifications?**

Reviewer #1: Major revision is recommended.

Reviewer #2: The paper appears to be good. But I think a few lines about ART and drug interactions need to be mentioned. Also tuberculosis is the most important opportunistic infection in India so a few lines on this treatment modality needs to be mentioned.

Suggestion Accept with minor revision.

Reviewer #3: (No Response)

**Summary and General Comments**

Reviewer #1: General comments:

This is an interesting manuscript that analyses outcomes, impact on equity, feasibility and acceptability of the treatment options in HIV co-infected VL patients. 

For this manuscript to have impact on policy, a few issues need to be transparently discussed and to revisit the aim of this manuscript. Given that no effective treatment exists for such patients, I wonder why the authors chose to carry out comparison of two treatment regimens: combo and monotherapy? The report will have more impact if it can highlight the critical gaps in therapeutic options for HIV co-infected VL patients rather than comparing two regimens that are both sub-optimal. As the NP stakeholders have identified relapse as the main feature of HIV-VL and the biggest challenge in treatment; valuations of outcome, impact on equity, feasibility and acceptability will be meaningful if there was a focus on the biggest challenge and how patients and stakeholders recognize the challenges and perceive public health constraints in this domain.

In a recent published report of a northern Ethiopia study, efficacy of total dose of 40mg/kg of ambisome monotherapy at day 29 was 50%, and the combination of Ambisome (30mg/kg total dose) and 28-day regimen of miltefosine resulted in 67% initial cure. At day 56 cure rate was 55% in the monotherapy arm, but 88% in the combo when treatment was extended by additional 28 days. Data on extended treatment is not available for the monotherapy, as it was not planned in the mentioned trial of northern Ethiopia. These findings show that the treatment of HIV co-infected patients will remain to be a challenge. Given that the monotherapy was horribly ineffective at the tested regimen, the comparative approach taken in this study does seem to be biased and in favour of the combination treatment obviously due to the better (but sub-optimal) efficacy. In the follow-up report of the above-mentioned study involving 54 patients, it was shown that the probability of relapse-free survival at one year was 50% (53% in 22 patients with baseline CD4 count > 200/µl hence not having secondary prophylaxis; and 46% in 29 patients with CD4 cells < 200/µl who were on secondary prophylaxis). It is important to avail such information to study participants (P & NP) to enable them to make a fair judgement and valuation. The fact that the NP stakeholders emphasized on the importance of the 6-month cure is indicative of the recognition of the challenges in treatment., i.e., the high risk of recurrent relapses and death. The premise (and context) for the qualitative and quantitative studies of patient outcomes should be clearer and well informed.

Specific comments

1. This study reveals that combination therapy was more feasible and acceptable than monotherapy mainly due to a shorter duration of hospitalization. Because the trial results are yet to be translated to policy, and the policy/guideline recommendations are yet to come; iIt is not yet clear how patients would be treated in the routine health settings using the combo or monotherapy to reveal which regimen has a shorter duration of hospitalization. The combo treatment regimen for South-East Asia and Eastern Africa regions are varied and necessitate the disaggregation of the data by geographic regions. In the case of the Eastern Africa region, the 28-day miltefosine in the combo (which will be 56 days when extended) is no shorter than the monotherapy that lasts 24 days (days 1,2,3,4,5,10, 17,24). In assessing acceptability, equity and feasibility comparing the combo with monotherapy; the tools (questionnaire) were not reinforced with specific questions that could help to substantiate claims on better acceptability, feasibility, and equity of the combo treatment regimen. A second layer of questions should have been included why P/NP study participants believe a treatment regimen was more acceptable, more feasible, and more equitable. It would have been preferred if context was projected, and explicit information why combo was found to be more equitable, more acceptable, and more feasible was interrogated. Furthermore, looking carefully at the narrations of the results, equity was not an issue either in the combo or monotherapy. Similarly, concerning feasibility, the only issue was the contraindication in pregnant women. Prolonged hospitalization could be an issue of feasibility and acceptability, but only in South-East India. As mentioned above, the crux of the investigations in this report would preferably focus on the management of recurrent relapses, i.e., the need for management of the chronic conditions and quality of life including the need for secondary prophylaxis. To my opinion, the authors did not demonstrate the existence of consensus to support the claims of feasibility, equity and acceptance of the combo more than the monotherapy. It is important to recognize that even with the best treatment options available today, the expectancy of survival in HIV co-infected patients is short (3 years at maximum). Given this, the emphasis on assessing quality of life of patients would have helped to advocate for further studies on new approaches and avenues of intervention to improve quality of life and to prolong years of survival - as in the case of ARTs in HIV/AIDS patients. 

2. Table 1 gives 26 females. Does it mean the number of males was 151? The table does not give the figure for males.

3. Figure 3 is presented in a biased way favouring the combo or focussing on combo (in a non-comparative way). For instance, the figure on acceptability is exclusive to combo (not showing rates for monotherapy). Only the feasibility component is presented in a comparative way. It is difficult to see from the figure the valuations about the monotherapy concerning acceptability and equity.

4. Figures 1,2, 3 lack title and legend.

Reviewer #2: The paper appears to be good. But I think a few lines about ART and drug interactions need to be mentioned. Also tuberculosis is the most important opportunistic infection in India so a few lines on this treatment modality needs to be mentioned.

Suggestion Accept with minor revision.

Reviewer #3: (No Response)

PLOS authors have the option to publish the peer review history of their article (what does this mean?). If published, this will include your full peer review and any attached files.

Reviewer #1: No

Reviewer #2: No

Reviewer #3: No
---

## [Decision Letter · Decision Letter 1]

13 Mar 2022

Dear Dr. Akl,

Thank you very much for submitting your manuscript "Stakeholders’ views and perspectives on treatments of visceral leishmaniasis and their outcomes in HIV-coinfected patients in East Africa and South-East Asia: a mixed methods study" for consideration at PLOS Neglected Tropical Diseases. As with all papers reviewed by the journal, your manuscript was reviewed by members of the editorial board and by several independent reviewers. The reviewers appreciated the attention to an important topic. Based on the reviews, we are likely to accept this manuscript for publication, providing that you modify the manuscript according to the review recommendations. 

Sincerely,

Helen P Price, PhD

Deputy Editor

Helen Price

Deputy Editor

Reviewer's Responses to Questions

**Key Review Criteria Required for Acceptance?**

**Methods**

-Are the objectives of the study clearly articulated with a clear testable hypothesis stated?

-Is the study design appropriate to address the stated objectives?

-Is the population clearly described and appropriate for the hypothesis being tested?

-Is the sample size sufficient to ensure adequate power to address the hypothesis being tested?

-Were correct statistical analysis used to support conclusions?

-Are there concerns about ethical or regulatory requirements being met?

Reviewer #1: The aim of the study still is unclear.

Reviewer #3: Second Review of PNTD-D-21-01369-R1

The responses below cover the author responses to Reviewer 3:

• The difference of participation from the two regions still needs further description, as does the implication on the overall validity of the study. It seems from the author reply that there were 32 stakeholders responded; however they then go on to describe 177 completed responses being received – this is confusing. It would important to be clear on the number of participants from each region at each stage of the study – please add absolute numbers to the percentages.

• The authors are correct regarding what is written in the WHO technical report series 949; however it is common knowledge that this is in conflict with WHO HIV guidelines. Considering the scope of 949 is VL, written by a panel of VL experts (not HIV) it is of minimal relevance with regards to classification of VL as an AIDS defining condition since national programme HIV guidelines (and decisions to initiate ART) are based on the WHO HIV guidelines alone, and not the 949 report. This is reflected in all national HIV guidelines. This is well discussed in the following paper, and I think it is important to state this as this is a WHO-authored manuscript:

van Griensven J, Ritmeijer K, Lynen L, Diro E (2014) Visceral Leishmaniasis as an AIDS Defining Condition: Towards Consistency across WHO Guidelines. PLoS Negl Trop Dis 8(7): e2916. https://doi.org/10.1371/journal.pntd.0002916

• ‘We agree with the Reviewer that both infections occur at low CD4 counts. The sentence meant to highlight that L. infantum occurs at a lower CD4+ count as compared to L. donovani (according to the following reference: https://www.who.int/leishmaniasis/resources/Leishmaniasis_hiv_coinfection5.pdf?ua=1, “Relapses due to L. donovani occur at higher CD4+ counts than is the case with L. infantum in Europe.”)’

The reference refers to relapses, not primary infections. This needs to be corrected or removed.

• "In this paragraph, we are describing the standard recommended regimens as described in the WHO Technical Report Series 949 (https://apps.who.int/iris/handle/10665/44412). However, we agree with the Reviewer that the duration of admission or further treatment depends upon a case-by-case basis and upon a variety of factors, and have added this information at the end of this paragraph"

The 949 report does not include the 14-day regimen; this was first described by Mahajan et al in 2015. The 949 report was written in 2010. Same applies for the E African regimen. The 949 report on page 66 describes only one regimen: Lipid formulations infused at a dose of 3–5 mg/kg daily or intermittently for 10 doses (days 1–5, 10, 17, 24, 31 and 38) up to a total dose of 40 mg/kg are recommended (A). My point is that the East African regimen is repeated as per their publication in the majority of cases; and their conclusion recommends this. 

As a side point, I believe the South Asia study has now been published so should be referenced https://pubmed.ncbi.nlm.nih.gov/35147680/

• The authors state in the reply that interviews conducted with patients were performed by telephone or using an on-line platform. Specifically, were the telephone calls with patients recorded (the authors seem to confirm this was the case later on in the responses, but it is important to be explicit here) and how were thematics and answers coded? Normally interviews require quite a lot of transcribing, how was this done and were any interviews double-coded? I can imagine that conducting this sort of interview over the phone must have been exceptionally challenging, and this should be explored in the text.

• "In addition, we would like to clarify that we are aware that in South Asia, e.g., India, the number of stakeholders is high due to the presence an elimination programme. This could explain why we received most email addresses and a massive response for survey participation from this region."

I am not sure I agree with this argument. Regardless of the elimination programme, the numbers of individuals involved with management and treatment of VL-HIV patients in India is extremely small – there are basically two centres that treat these patients (Kolkata and RMRI, Patna), who have been treating patients for the last 5 years or so. On the contrary, there is a vast range of decades worth of experience of VL/HIV in Ethiopia and Sudan, and as you can see from the attendance of meetings and consortiums (take ASCEND recently, for example) there are a wide range of stakeholders in the East African setting ranging from NGO, governmental, academic and beyond. Information and sources are not rudimentary; there are clear individuals and organisations that could have been approached to get a wider buy in when it became obvious that the initial sample framework for East Africa was inadequate. I think it is incorrect for the authors to explain this in this way; it seems rather that there was a greater effort made (unintentionally perhaps) for South Asia than for East Africa. This is important to self-reflect on, as it cuts to the core and validity of the messages and outcomes. 

• “Wherever possible, attempts were made to contact all VL patients under trial by the treating physicians. Patients were given information about this study, and were invited for the interview. Local principal investigators (DP and NM) then contacted patients who volunteered and expressed their willingness, obtained their informed consent and conducted the interviews.”

So if I understand correctly, all the VL-HIV patients were those that had previously participated in the phase 3 clinical trial on which the guideline change is based? This needs to be made explicit in the manuscript, and median time from treatment reported as the potential impact of recall bias should be clear, since this study was conducted long after the conclusion of the clinical trials. If correct, this is both a strength and a weakness for the study. A strength as this reflects directly patients who had either one of the treatments – and it will be critical to describe the proportions of those interviewed who received which treatment (eg. If all the patients received one of the treatment arms, they would not be a meaningful qualitative comparison). On the other side, all of these patients received treatment under clinical trial conditions, as such were admitted for the duration of treatment in hospital and receiving a level of care that would not be seen under routine conditions. These should be discussed in the limitations.

• "The terminology “beneficiary impact perception” is not clear to us. We appreciate clarification so we could address it appropriately"

Beneficiary impact perception is the perspective of the beneficiary (i.e. the patient with VL-HIV). I mean the purpose of interviewing this group was to determine their perspectives on the treatments they received – and the impact of the treatment on them. This is a much under-explored area in clinical trial research, and I commend the authors for exploring this – but they should try and go deeper into this as it is really important, especially as the number of patients (19, I believe) is really a huge sample size for such work. The authors state that there was insufficient information from the patient’s perspective – this simply cannot be the case with such a sample size unless there was something fundamentally wrong or lacking with the interview methodology used. If this is the case, that is fine, but it needs to be explained in the text.

• "We agree that snowball sampling would have been a great approach, especially for the qualitative part of our study (interviews). As we already had the survey as our gateway to recruitment for the qualitative study, and as it resulted in a good number of interviewees, we did not need to resort to snowball sampling. Had the survey not yielded a good number of interviewees, we might have resorted to this sampling strategy."

I find this response confusing still. The authors accept that there were insufficient responses from the East African region, but then state here that a ‘good number of interviewees’ were identified… Clearly, the method of participant sampling from East Africa was inadequate as it failed, so this should be expressed/explained.

**Results**

-Does the analysis presented match the analysis plan?

-Are the results clearly and completely presented?

-Are the figures (Tables, Images) of sufficient quality for clarity?

Reviewer #1: Yes, results are clear. But one aspect of the investigations, i.e., the comparison of treatment duration, was carried out with a wrong premise. See general comments section below.

Reviewer #3: (No Response)

**Conclusions**

-Are the conclusions supported by the data presented?

-Are the limitations of analysis clearly described?

-Do the authors discuss how these data can be helpful to advance our understanding of the topic under study?

-Is public health relevance addressed?

Reviewer #1: The conclusion is unclear. See general comments section below.

Reviewer #3: (No Response)

**Editorial and Data Presentation Modifications?**

Reviewer #1: Please refer general comments section below..

Reviewer #3: (No Response)

**Summary and General Comments**

Reviewer #1: The manuscript has been improved substantially and given us a placated understanding of the problems in the treatment of HIV co-infected VL patients. However, I would like to invite the authors to address following: 

Concerning the AmBisome monotherapy treatment regimen, the authors have referred us to the WHO Technical Report Series 949, which indeed indicates treatments on days 1-5, 10,17,24, 31 and 38. This regimen has not been used in Eastern Africa endemic region, and the recommendation itself was very general and makes no reference to the endemic regions. At the time of the publication of Serie 949, no RCTs were conducted for VL patients with HIV co-infection. 

In the recent RCTs conducted in Eastern Africa (mainly Ethiopia), the miltefosine and AmBisome combo was used for minimum 28 days, and up to 56 days while monotherapy at 40mg/kg maximum dose was given in a treatment regimen lasting 24 days (days 1-5, 10, 17, and 24). It looks like the authors were not aware of this information, which had led them believe Ambisome monotherapy required longer duration of treatment. One then wonders if study participants were given the right information about treatment duration concerning HIV co-infection of VL. In one of the annexes/excel file/, responses like “Two drug is more feasible as it will require short duration of hospital stay” has been quoted. It is implied that use of two drugs will have the advantage of shorter hospital stay/treatment duration, which in fact is true in many conditions. However, as mentioned above the monotherapy had shorter duration contrary to what the authors considered. Thus, the section of this manuscript that dealt with duration of treatment is not valid.

Further, the RCTs in Ethiopia have demonstrated poor efficacy of AmBisome monotherapy. Again, one wonders why this article we are reviewing had to deal with comparison of the inefficacious treatment with the combo treatment. In my original review, I asked what the aim of the study was concerning treatment options of HIV co-infected VL patients. I also raised a query about the lack of conclusions. The whole issue concerning feasibility, acceptability, valuation of outcomes and impact on equity makes sense if the interventions in place fulfil a minimum set of criteria related to patient outcomes (cure, relapse free state, improved quality of life). If I may ask the same question again, why was this study conducted given the fact that none of currently available treatments are effective enough? Is it because both regimens (mono and combo) are inefficacious, and that one needs to choose the better of these badly performing treatments?

Reviewer #3: The answers are generally ok, but honestly it feels like the referencing is quite sloppy, and the underlying knowledge of the subject material limited, which is surprising since the author list includes some of the major players in VL. This sort of study -and I really do refer to the patient perspectives rather than the non-patient perspectives (which one could probably guess the answers and in my opinion reveals nothing novel), is critically important and I feel that the authors have really under-sold the study by not focusing much more on the outcomes of the patient interviews, or at least explaining why they were not able to generate decent evidence.

More generally, I think the authors need to be clear and transparent on the failure to include adequate sampling from East Africa, and how that impacts the validity of the results.

PLOS authors have the option to publish the peer review history of their article (what does this mean?). If published, this will include your full peer review and any attached files.

Reviewer #1: No

Reviewer #3: No

Figure Files:

Data Requirements:

Reproducibility:

References

---

## [Decision Letter · Decision Letter 2]

30 May 2022

Dear Dr. Akl,

Thank you very much for submitting your manuscript "Stakeholders’ views and perspectives on treatments of visceral leishmaniasis and their outcomes in HIV-coinfected patients in East Africa and South-East Asia: a mixed methods study" for consideration at PLOS Neglected Tropical Diseases. As with all papers reviewed by the journal, your manuscript was reviewed by members of the editorial board and by several independent reviewers. The reviewers appreciated the attention to an important topic. Based on the reviews, we are likely to accept this manuscript for publication, providing that you modify the manuscript according to the review recommendations. 

Sincerely,

Mitali Chatterjee

Associate Editor

Helen Price

Deputy Editor

Reviewer's Responses to Questions

**Key Review Criteria Required for Acceptance?**

**Methods**

-Are the objectives of the study clearly articulated with a clear testable hypothesis stated?

-Is the study design appropriate to address the stated objectives?

-Is the population clearly described and appropriate for the hypothesis being tested?

-Is the sample size sufficient to ensure adequate power to address the hypothesis being tested?

-Were correct statistical analysis used to support conclusions?

-Are there concerns about ethical or regulatory requirements being met?

Reviewer #3: (No Response)

**Results**

-Does the analysis presented match the analysis plan?

-Are the results clearly and completely presented?

-Are the figures (Tables, Images) of sufficient quality for clarity?

Reviewer #3: (No Response)

**Conclusions**

-Are the conclusions supported by the data presented?

-Are the limitations of analysis clearly described?

-Do the authors discuss how these data can be helpful to advance our understanding of the topic under study?

-Is public health relevance addressed?

Reviewer #3: (No Response)

**Editorial and Data Presentation Modifications?**

Reviewer #3: (No Response)

**Summary and General Comments**

Reviewer #3: Thank you to the authors, the manuscript is much improved and acceptable for publication with the following slight adjustments:

1) Participation from East Africa:

The authors still insist that the issue was of uptake; this is simply not true. The issue was that only 24% of invitations to participate were to East African stakeholders. In its worst interpretation, the current wording could be interpreted as a lack of interest from East African stakeholders, which is untrue (and honestly abit offensive) when this was rather a failure of methodology, and should be recognised as such. I suggest changing the limitation sentences to:

‘’Another limitation is that this study invited lower participation from East Africa as compared to South-East Asia.”

AND

“As we used the survey as our gateway to recruitment for the qualitative study, the substantially fewer invitations sent to East African participants in the survey study also reflected on the interviews

As a side note - ASCEND is a much bigger and successful programme in East Africa, where there are far more than one representative per country. Again, this response suggests an overly South Asian focus from the researchers..

2) Patients responses:

Suggest making this more explicit and better worded:

Change:

Physicians at the two treating centres in Ethiopia and India where the clinical trials were conducted (7, 8) helped with patient recruitment, while ensuring voluntary participation, confidentiality and anonymity. Wherever possible, attempts were made to contact all VL patients under trial by the treating physicians.”

To:

“At the two treatment centres where the clinical trials were conducted, physicians attempted to contact all the patients who had participated in the trials, while ensuring voluntary participation, confidentiality and anonymity,”

3) Recall Bias:

I disagree with the authors on this point. Even the patients recollection of their experiences with the disease will be impacted by recall bias, and its important from a comparison perspective also; eg. If the median duration was 5 years in Ethiopia vs 6 months in India, it is important to present this so appropriate interpretation on the integrity of the responses can be made. To this effect, please include the median and IQRs of the time-from-treatment to interview for each region.

With regards to treatment, I find it somewhat unrealistic that the physicians were able to explain the ‘other’ treatment to the patients, and then elicit a meaningful opinion from them on which one they feel is better. Its also not possible to detach patient’s attitudes towards the treatments they received and the way they received it – so the fact that they were all treated as inpatients in both countries is important, and is a limitation on the validity of these perspectives in the real world setting where ambulatory treatment is likely to be encouraged, and where the standard of care/follow up/support is going to be far lower in the non-clinical trial context. 

My suggestion is to briefly expand on this area; the methodology that the authors have used is extremely valuable and rare in NTD research, and all learning points and evidence should be presented accordingly – as I said in my previous review, the patient feedback is by far the most interesting and novel component of this research.

4) ‘Openness more in telephone interviews’

This concept is surprising to me, and it would be good if the authors can provide some references to support this. In my understanding, for sensitive matters that are significant events, participants feel much more comfortable being interviewed in person, in private, where both a rapport can be built and privacy can be guaranteed. Typically, this done by inviting participants to attend a ‘safe’ place of interview separate to the residence (with travel and wage loss compensation paid). Over the phone, there is no guarantee of privacy, as the interviewer does not know the situation of the interviewee who may not be in a position to express that they are able to talk freely. 

I would suggest that the statement “On that note, the use of telephone interviews further conserved confidentiality and anonymity of patients” is removed from the manuscript unless there is clear evidence to support this supposition. I suspect that the reason why patients were not open on the telephone was precisely because it is very difficult to build a rapport over the phone for sensitive subjects, and this should be recognized in the manuscript text as a potential limitation/cause of the poor result yield.

PLOS authors have the option to publish the peer review history of their article (what does this mean?). If published, this will include your full peer review and any attached files.

Reviewer #3: No

Figure Files:

Data Requirements:

Reproducibility:

References

---

## [Editor Report · Decision Letter 3]

29 Jun 2022

Dear Dr. Akl,

We are pleased to inform you that your manuscript 'Stakeholders’ views and perspectives on treatments of visceral leishmaniasis and their outcomes in HIV-coinfected patients in East Africa and South-East Asia: a mixed methods study' has been provisionally accepted for publication in PLOS Neglected Tropical Diseases.

Best regards,

Mitali Chatterjee

Associate Editor

Helen Price

Deputy Editor

---

## [Editor Report · Acceptance letter]

11 Aug 2022

Dear Dr. Akl,

We are delighted to inform you that your manuscript, "Stakeholders’ views and perspectives on treatments of visceral leishmaniasis and their outcomes in HIV-coinfected patients in East Africa and South-East Asia: a mixed methods study," has been formally accepted for publication in PLOS Neglected Tropical Diseases.

Best regards,

Shaden Kamhawi

co-Editor-in-Chief

Paul Brindley

co-Editor-in-Chief
